# INSurVeyor: improving insertion calling from short read sequencing data

Ramesh Rajaby[1,2], Dong-Xu Liu[3,4], Chun Hang Au [1], Yuen-Ting Cheung [1], Amy Yuet Ting Lau[1], Qing-Yong Yang [3,4] & Wing-Kin Sung [1,2,3,4,5,6,7] ✉

Insertions are one of the major types of structural variations and are defined as the addition of 50 nucleotides or more into a DNA sequence. Several methods exist to detect insertions from next-generation sequencing short read data, but they generally have low sensitivity. Our contribution is two-fold. First, we introduce INSurVeyor, a fast, sensitive and precise method that detects insertions from next-generation sequencing paired-end data. Using publicly available benchmark datasets (both human and non-human), we show that INSurVeyor is not only more sensitive than any individual caller we tested, but also more sensitive than all of them combined. Furthermore, for most types of insertions, INSurVeyor is almost as sensitive as long reads callers. Second, we provide state-of-the-art catalogues of insertions for 1047 *Arabidopsis Thaliana* genomes from the 1001 Genomes Project and 3202 human genomes from the 1000 Genomes Project, both generated with INSurVeyor. We show that they are more complete and precise than existing resources, and important insertions are missed by existing methods.

Structural variations (SV) are defined as variations in a genome involving 50 base pairs or more. Although less frequent than point mutations and short indels, they account for more heritable differences in the population[1]. Insertions are a major type of SV, and they are defined as the introduction of a sequence of 50 nucleotides or more into a locus in a DNA segment. The extra sequence is known as the inserted sequence and the locus is known as the insertion site.

Aside from being a major source of polymorphism in the population, insertions have been observed to be involved in several diseases. Examples include diseases of the central nervous system[2], hemophilia[3], cancers such as colon[4], colorectal[5], gastrointestinal[6], several neurodegenerative disorders[7–11] and many others[12].

Nowadays, insertions are usually detected using either long read sequencing technologies (like PacBio HiFi reads and ONT nanopore reads) or paired-end short read sequencing technologies (like Illumina reads). Long read sequencing allows for substantially more accurate and sensitive insertion detection. However, it is still expensive, which limits its use in population studies. Another solution is to use paired-end short reads. Sequencing of short reads datasets is nowadays inexpensive and for this reason it is the technology of choice for sequencing large populations. The number of short reads datasets available is increasing exponentially, with studies on tens of thousands of genomes being published[13,14]. With projects such as the Hong Kong Genome Project, Singapore SG100K, the European '1+ Million Genomes' Initiative and the All of Us Research Project, short-read WGS sequencing will remain at the center of genomics for the foreseeable future.

Calling insertions from short reads is difficult. In principle, SV callers should be able to detect them. A large number of such methods exist. Cameron et al.[15] published a comprehensive benchmarking of

[1]Hong Kong Genome Institute, Hong Kong Science Park, Shatin, Hong Kong, China. [2]A*STAR Genome Institute of Singapore, 60 Biopolis Street, Singapore 138672, Singapore. [3]National Key Laboratory of Crop Genetic Improvement, College of Informatics, Huazhong Agricultural University, Wuhan 430070, China. [4]Hubei Key Laboratory of Agricultural Bioinformatics, College of Informatics, Huazhong Agricultural University, Wuhan 430070, China. [5]Department of Chemical Pathology, The Chinese University of Hong Kong, Hong Kong, China. [6]Laboratory of Computational Genomics, Li Ka Shing Institute of Health Sciences, The Chinese University of Hong Kong, Hong Kong, China. [7]School of Computing, National University of Singapore, 13 Computing Drive, Singapore 117417, Singapore. ✉e-mail: kwksung@cuhk.edu.hk

the existing SV callers, and found GRIDSS[16] and Manta[17] to have the best performance, followed by Delly[18] and Lumpy[19]. However, not all SV callers explicitly report insertions, and when they do the sensitivity is low[20]. When applied to recent comprehensive benchmark catalogues of SVs, all the callers we tested consistently detected <40% of the insertions (shown below).

Aside from general callers, several specialised methods that target specific groups of insertions have been published over the years. Insertions of mobile elements such as Alu and L1 represent a well known cause of genetic variability in the population[21,22], and tailored methods such as MELT[23], Mobster[24], xTea[25] and others[26] have been created. Methods such as Pamir[27] and PopIns2[28] have been designed to detect novel insertions, i.e., insertions whose sequence is not present in the reference genome. TranSurVeyor[29] targets insertions due to transposition. However, these methods miss many of the insertions they were designed for, as we later show in our experiments. Furthermore, several meta-callers have been developed, such as MetaSV[30] and Parliament2[31], which integrate the output of multiple callers in order to increase recall. However, as shown below, combining multiple callers results in a higher runtime and there is a sharp diminishing return for the increase in recall as more methods are added. Another drawback is an increased number of false positives[20].

A major challenge in detecting insertions from short reads is that, for many insertions, the inserted sequence (or part of it) is similar to multiple regions in the reference genome. This causes reads related to the same insertion to align to different locations. Because existing callers fail to realise that these reads represent the same insertion, they introduce false positives and miss many true positives. Previously, we tackled the issue by employing a strategy based on multiple sequence alignment to identify the source of the insertion in the reference[29]. Unfortunately, this approach is limited to insertions due to transposition, i.e. when the inserted sequence is identical (or nearly identical) to a region of the reference genome. We found that a large portion of the inserted sequences are not fully present in the reference. Even when they are, the most similar copy in the reference may be substantially different. In particular, we found that only 22% of the inserted sequences in the HG002 benchmark and 44% of the inserted sequences in the HGSVC2 dataset are accurately represented in the reference genome. For this reason, many insertions are not reported, or an incorrect or imprecise inserted sequence is reported.

In this work, we aim to solve this problem and substantially fill the gap between short and long reads. For this purpose, we introduce INSurVeyor, a method that detects insertions from paired-end WGS data. INSurVeyor addresses the aforementioned issues by using three distinct algorithms: (i) when a region in the reference is sufficiently similar to the inserted sequence, INSurVeyor identifies the region and uses a reference-guided assembly algorithm to produce the exact inserted sequence; (ii) when the inserted sequence is not present in the reference, INSurVeyor employs an ad-hoc de-novo assembly algorithm, which uses information about the strands of the reads to remove wrong assemblies; and (iii) a specialised module detects smaller insertions. This allows INSurVeyor to predict any type of insertion and to report accurate inserted sequences. In the remainder of this article, we first show an overview of the method implemented in INSurVeyor. Then, we use several publicly available benchmark human genomes to show that INSurVeyor is not only more sensitive than any individual caller we tested, but also more sensitive than all of them combined. Our method predicts >1400 true positive insertions per sample that are missed by popular state-of-the-art methods, while maintaining excellent precision. Furthermore, it is more sensitive in predicting mobile elements insertions (MEI) than specialised, database-based MEI callers. With the exception of insertions of low complexity sequences, INSurVeyor achieves performance close to the state of the art in SV detection with long reads. We then demonstrate that INSurVeyor performs well on non-human genomes. Liu et al.[32]

recently published an ensemble SV caller, IndelEnsembler, and used it to study the SVs of a population of 1047 *Arabidopsis Thaliana*. We tested INSurVeyor on the benchmark plant genomes used in ref. 32 and show that it consistently outperforms IndelEnsembler. While IndelEnsembler detected on average 55% of the insertions in seven A. Thaliana benchmark datasets, INSurVeyor detected 85% of them, while maintaining similar or superior precision. We observed similar improvements when we tested the software on two more species of plants. We then proceeded to call insertions on the same 1047 A. Thaliana used by Liu et al., and found multiple previously missed insertions that significantly influence phenotypes such as flowering time, days until first open flower and rosette leaf number. Finally, we present a catalogue of insertions for the full 1000 Genomes Project dataset. Insertions were called by INSurVeyor on 3202 high-coverage human genomes in <3 days using modest resources. When compared to the current state of the art, our catalogue contains nearly three times as many insertions. We show that not only we predicted nearly all the previously detected insertions, but also 94,988 novel ones, with high validation rates. Furthermore, we show that INSurVeyor identified polymorphism in 567 potentially clinically relevant loci, mostly novel.

INSurVeyor is available at https://github.com/kensung-lab/INSurVeyor and is fully open source. It is easy to run and only requires an indexed BAM or CRAM file and a reference genome, without the need for additional data or annotations, and outputs a fully standard VCF file.

The dataset generated for this study are also available, and links are provided in Data availability.

## Results
### Overview of INSurVeyor
For many insertions, the inserted sequence is similar to multiple regions in the reference genome (mobile element insertion is a typical example), and reads related to those insertions can align to multiple locations. For this reason, most existing structural variation callers either fail to predict these insertions, or they call a large number of false positives[29].

TranSurVeyor[29], a tool specialised in the detection of insertions due to transposition, tackles the problem by identifying read pairs that support the presence of an insertion, and labels one read as *stable* and the other as *unstable*. The stable read is presumed to align to the genomic region surrounding the insertion site, while the unstable read is presumed to be sequenced from the inserted sequence. Stable reads are clustered to identify insertion locations and, for each cluster, the unstable reads are aligned to the reference using a multiple sequence alignment heuristic algorithm, to determine the source of the transposed sequence.

However, this technique has limitations, as it can only detect an insertion if its inserted sequence is highly homologous to a segment of the reference genome. Using high-quality structural variations catalogues we confirmed that this is not the case for most insertions. In particular, we observed that only 22% of the inserted sequences in the Genome in a Bottle (GIAB) HG002 SV benchmark[33] and 44% of the inserted sequences in the HGSVC2 catalogue[34] are accurately represented in the reference genome (Supplementary Fig. 1). This causes many insertions to be missed or reported with an inaccurate inserted sequence.

Our method addresses these shortcomings in two ways. When the inserted sequence is similar to a region in the reference, INSurVeyor uses reference-guided assembly to determine the inserted sequence, instead of simply reporting the reference region. Therefore, the precise inserted sequence is provided. When no sufficiently similar or complete reference region is found, INSurVeyor performs de novo assembly of the stable and unstable reads associated with the insertion; the assembly algorithm takes into account information about the mapping location and the orientation of the stable reads to improve

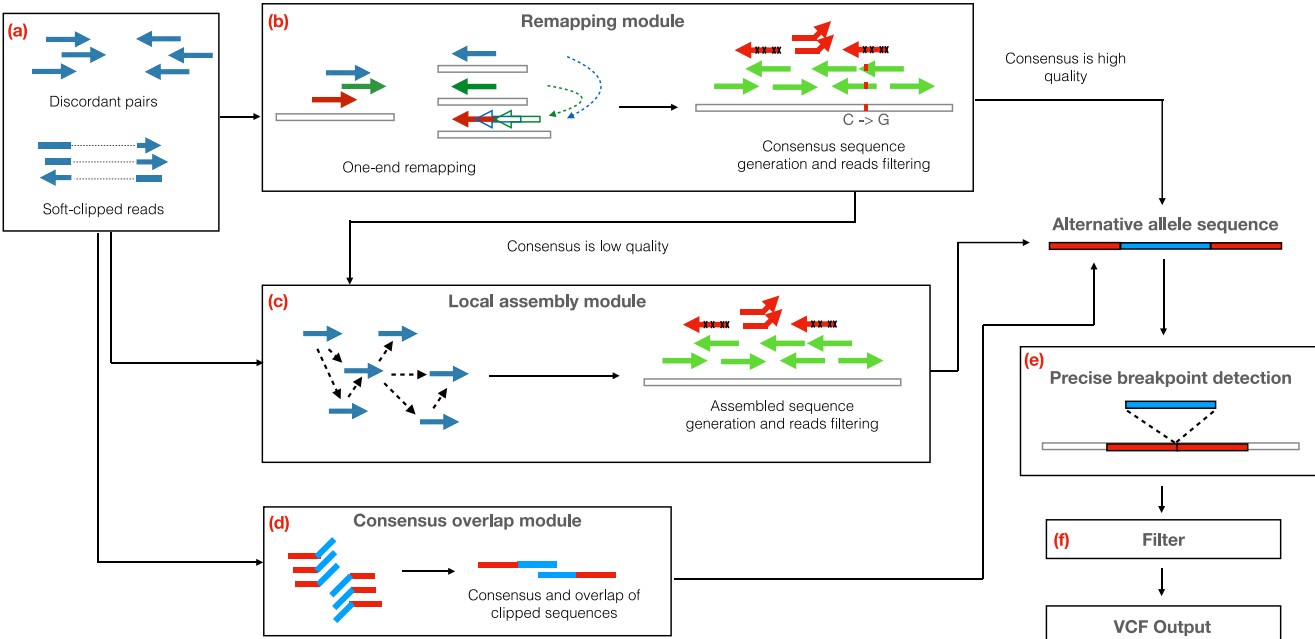

**Fig. 1 | Overview of the INSurVeyor method.** The method can be essentially divided into three blocks: (**a**) INSurVeyor extracts discordant pairs and clipped reads as possible evidence of insertions; (**b–d**) the evidence extracted by (**a**) is used to generate the alternative allele sequence, which consists of the predicted inserted sequence along with the two flanking regions shared with the reference genome. This is achieved by three separate modules: the remapping module (**b**) aims at predicting transpositions; the local assembly module (**c**) aims at predicting novel insertions, while the consensus overlap module (**d**) predicts small insertions. **e** This sequence is then remapped to the reference genome to identify the precise boundaries of the predicted insertion, which is finally passed through a series of filters (**f**) that aim at reducing the number of false positive calls.

the precision of the assembled sequences. Finally, short insertions may be missed because they are supported by few or no read pairs; a specialised consensus-overlap algorithm has been introduced to predict them. The result is INSurVeyor, a fast, sensitive and precise tool that is able to detect any type of insertion. In the remainder of this section, we provide a high-level overview of the algorithm.

INSurVeyor identifies insertions using three major steps. The first step extracts the subset of read pairs that is relevant for predicting insertions (Fig. 1a).

The second step aims at building, for each insertion, the sequence of the alternative allele, which consists of the inserted sequence and two flanking regions from the reference. This step is the core of INSurVeyor, and it is composed of three different modules that target insertions based on their characteristics: the *remapping module* (Fig. 1b) targets insertions due to transposition, i.e., when a sequence similar to the inserted sequence can be found in the reference genome; the *local assembly module* (Fig. 1c) targets novel insertions, i.e. insertions whose inserted sequences are not present in the reference genome. Finally, the *consensus-overlap module* (Fig. 1d) aims at predicting smaller insertions that may have been missed by the other two modules due to lack of supporting read pairs. Once the alternative allele is assembled, it is remapped to the putative insertion site in order to determine the precise breakpoints and the inserted sequence (Fig. 1e). Finally, the candidate insertions that pass a series of filters are reported (Fig. 1f). A detailed, technical explanation of each module can be found in Supplementary Information.

## HG002 benchmark

The Genome in a Bottle Consortium developed a benchmark catalogue of 7281 sequence-resolved insertions and 5,464 deletions for HG002[33], a genome in the Personal Genome Project. The benchmark, based on the human genome version 19 (hg19), was obtained by integrating different methods and technologies. Using the method described in Methods, 3273 out of the 7281 reported insertions were classified as

tandem duplications. These are outside of the scope of the methods tested here, so we excluded them. In the rest of the manuscript, we will refer to this catalogue as GIAB-SV.

The authors in ref. 33 defined a set of Tier 1 regions in hg19 where the callset is guaranteed to be reasonably complete, and any extra insertion predicted by the tested tools is very likely to be a false positive. Only predicted calls in Tier 1 regions are used for estimating precision. We obtained a BAM file of ~50× coverage by downloading runs from SRR1766442 to SRR1766486, and mapping them to hg19 using BWA-MEM[35].

We tested several tools, and chose the best representatives, to our knowledge, from three categories: general SV callers (Manta and Delly), theoretically able to predict any type of insertion; mobile element insertion (MEI) callers (MELT and xTea); and a novel insertion caller (Pamir). Details of the additional software tested and the criteria for selecting the callers are reported in Methods. Some tested tools do not report the actual inserted sequences, so benchmark and predicted calls are matched based on insertion site positions. For tools that report the inserted sequences (Delly, Manta, Pamir, INSurVeyor), the comparison is extended to assess the accuracy of the predicted inserted sequences. Technical details are reported in Methods.

INSurVeyor had much higher sensitivity than general SV callers, Delly (24×) and Manta (2×), as shown in Fig. 2a. Although sensitivity is also much higher than MELT (3×), Pamir (6.5×) and xTea (5×), they are not directly comparable as these are specialised tools that only aim at detecting a subset of the insertions. Precision was extremely high (0.98, second only to Delly), but all tools had good precision with the exception of Pamir. Even when pooling the calls from all the tools, INSurVeyor is still considerably more sensitive (2878 true positives for INSurVeyor and 2434 true positives for all the other tools combined, Fig. 2b). Furthermore, running all tools consecutively took >1200 min, while INSurVeyor only took 75 min. Clearly, adding more tools may increase the sensitivity, but at the expense of running time and possibly a higher number of false positives. The combination of existing

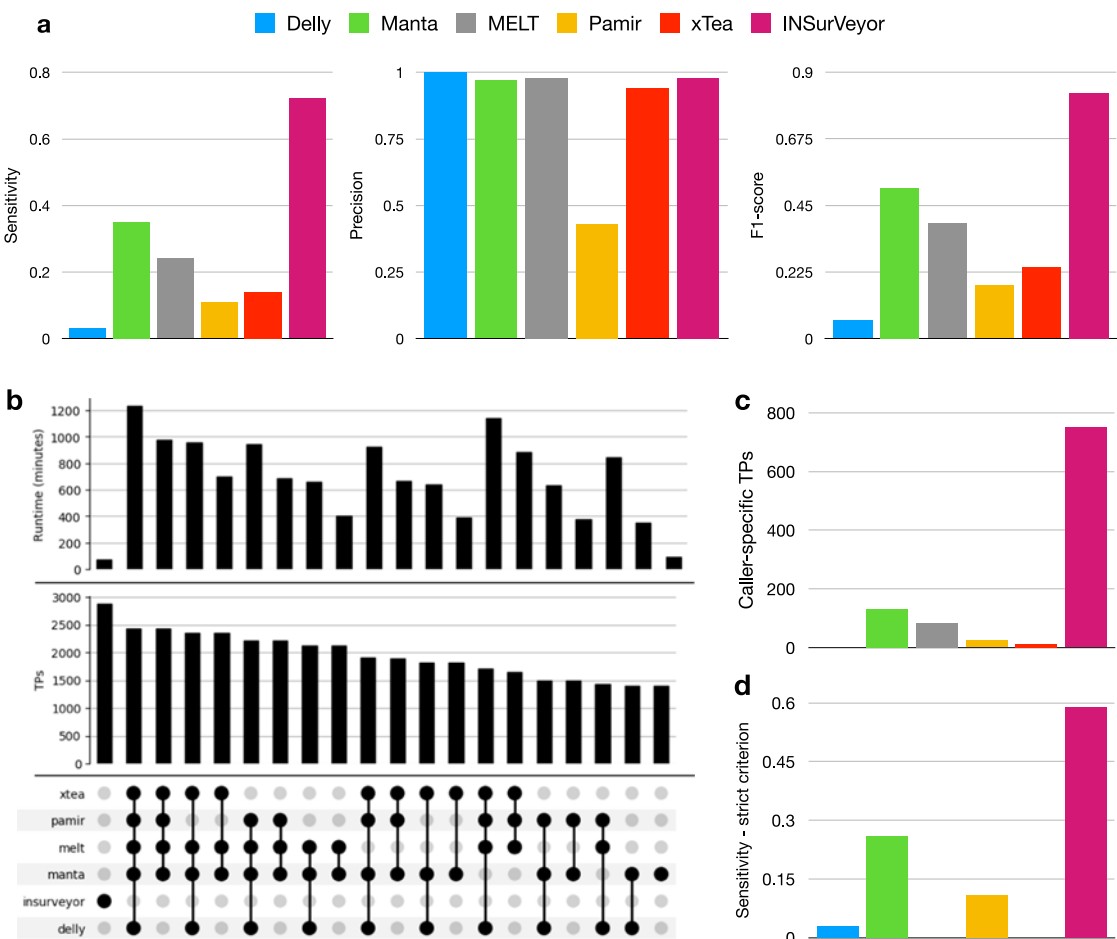

**Fig. 2 | Performance of the tested tools on the HG002 benchmark. a** Sensitivity, precision and F1-score of individual callers. INSurVeyor has a much higher sensitivity (0.72) than the other tools, and extremely high precision (0.98). This predictably results in the highest F1-score (0.83). **b** The number of predicted TPs and the running time in minutes for INSurVeyor and for different combinations of existing tools (sorted by number of TPs, top 20 showed). INSurVeyor alone predicts more true positives than all the other tools combined, while using a fraction of the running time. **c** The number of calls that are uniquely contributed by each caller. Notably, INSurVeyor contributes more than 700 true positives that are missed by all of the other tools. No other tested caller performs similarly. **d** Sensitivity when using the strict criterion. INSurVeyor is still more sensitive than other tools.

tools that seems to offer the best balance between running time and sensitivity is MELT and Manta. One or both of these tools are regularly used in large scale studies such as the 1000 genomes project[21,36], and GnomAD[13]. Even so, Manta and MELT collectively took 5 times longer than INSurVeyor to run (405 min) while producing less true positive calls (2126 for Manta and MELT, 2878 for INSurVeyor). Adding more tools provides little benefit in terms of discovered insertions, while it considerably increases the running time. Most importantly, nearly 800 insertions are discovered only by INSurVeyor and missed by all the other tools (Fig. 2c). Even when using the strict comparison criterion, which requires not only the insertion site but also the inserted sequence to agree with the benchmark (Fig. 2d), INSurVeyor is still more than twice as sensitive as Manta. The loss of sensitivity is almost entirely due to very long inserted sequences that INSurVeyor could only partially assemble; 97% of the completely assembled sequences agreed with the benchmark.

We further investigated how (a) the tools perform on different types of insertions, and (b) how our solution applied to short reads compare to the state of the art in long reads insertion calling. We downloaded a 50x PacBio HiFi dataset for HG002 (PRJNA586863) and ran Sniffles2[37], SVIM[38] and cuteSV[39], three popular SV callers for long reads. Sniffles2 had a slightly higher recall than the other two, so we chose it for our comparison. We partitioned the insertions into three major categories, according to their inserted sequences: mobile

elements (SINE and LINE, 1282, 32%), low complexity sequences (1111, 27.7%) and others (1615, 40.3%). Details on the classifications are reported in Methods.

INSurVeyor is more sensitive than the other tools that use short reads for every type of inserted sequence (Fig. 3). Note that MEI callers rely on databases of known mobile elements, while INSurVeyor does not use any prior knowledge, and is completely agnostic to the species analysed. Despite this, INSurVeyor has clearly superior sensitivity in detecting MEI (0.85 INSurVeyor, 0.61 MELT, Fig. 3a). Interestingly, MEI callers appear to perform especially poorly when a SINE element inserts into a reference SINE, or when a LINE element inserts into a reference LINE (Supplementary Fig. 2). Sniffles2 performs extremely well (0.99 sensitivity), as expected given the high quality of the long reads dataset provided. Despite this, INSurVeyor can detect >85% of MEI predicted using high fidelity long reads.

Insertions of low-complexity sequences are extremely challenging to detect using short reads, and they are the only type of insertion where using long reads provides a large advantage. This likely due to the highly repetitive nature of the sequences and other technical reasons, such as an usually large number of sequencing artefacts and possibly very low or very high GC content. Among the short reads callers, INSurVeyor shows a 32% increase in sensitivity over Manta, while the other tools are essentially unable to call such insertions (Fig. 3b).

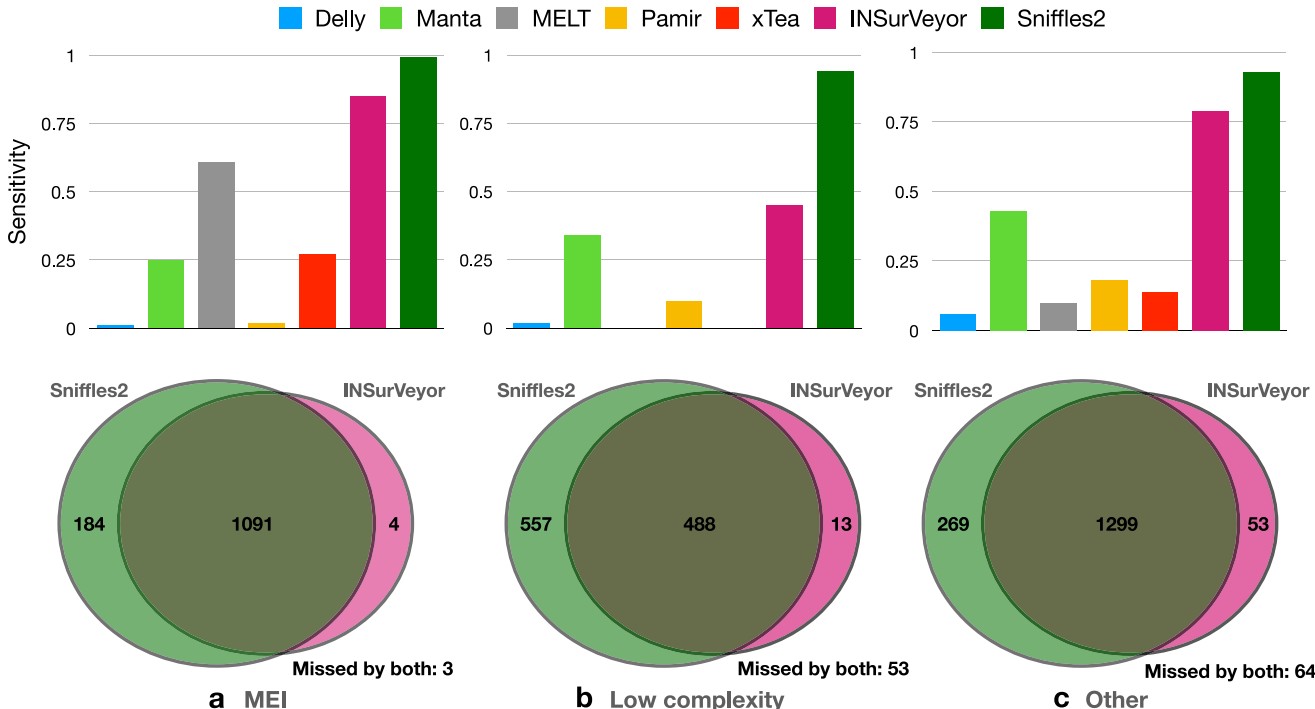

**Fig. 3 | Performance of the tested tools on different types of inserted sequences, and comparison with Sniffles2.** The benchmark insertions are partitioned into three types depending whether the inserted sequence is (**a**) a mobile element (SINE, LINE or SVA), **b** low complexity or (**c**) other, i.e., none of the previous categories. The sensitivity of different tools was assessed for each type. INSurVeyor performs better than other short reads-based methods in every single class. Furthermore, with the exception of low complexity sequences, INSurVeyor predicts most of the insertions detected by methods that use long reads datasets.

Finally, INSurVeyor showed the largest advantage on the insertions that were classified neither as MEI nor as low complexity insertions. Only Manta and INSurVeyor were able to call these insertions, but the latter was nearly twice as sensitive. Not only INSurVeyor was able to detect 82% of the insertions detected by Sniffles2, but it also detected a relatively large number of insertions missed by the long reads caller (Fig. 3c).

## HGSVC2 benchmark

The Human Genome Structural Variation Consortium (HGSVC) has published a catalogue (called HGSVC2) of insertions and deletions identified from long-read PacBio whole-genome sequencing and Strand-seq data for 34 human genomes from the 1000 genomes projects[34] (the catalogue also includes calls for a 35th genome, HG002, which we already covered). Furthermore, the New York Genome Center (NYGC) has recently performed whole-genome sequencing (WGS) of the 2504 original samples in the 1000 Genomes Project, along with 698 additional samples, on modern sequencing platforms and at high coverage (30×)[36]. We ran the callers on the CRAM files provided by the NYGC. Unfortunately, we failed to run xTea, and Pamir did not complete any sample within 1 week. Delly did not perform any better than it did on HG002, and we found little merit in including it in any subsequent analysis. Since Manta-MELT was also the most cost-effective combination on HG002, we decided to benchmark it against INSurVeyor.

When applied to the HGSVC2 samples, the results were similar to what was shown in the previous section. INSurVeyor is more sensitive than both tools combined and equally precise (Fig. 4a), which results in a higher F1-score (Fig. 4b). Remarkably, INSurVeyor predicts, on average, 1441 true positive insertions per sample that are missed by both Manta and MELT (compared to missing 251 on average, Fig. 4c).

## Arabidopsis thaliana

Next, we test INSurVeyor on a non-human genome. Recently, Liu et al.[32] developed an SV detection pipeline, IndelEnsembler, and used it to produce a state-of-the-art catalogue for 1047 *Arabidopsis thaliana* genomes. In order to detect insertions, IndelEnsembler combines calls from TranSurVeyor and Manta. Liu et al. generated a benchmark callset using Assemblytics[40] on seven publicly available assembled *Arabidopsis Thaliana* genomes. The performance assessment method employed in ref. 32 was applied to INSurVeyor.

Figure 5a reports sensitivity and precision for all seven samples. INSurVeyor greatly improves sensitivity over IndelEnsembler. Average sensitivity for IndelEnsembler was 0.55, compared to 0.85 for INSurVeyor. Furthermore, the increase in sensitivity does not come at the expense of precision. Liu et al. also tested IndelEnsembler on different plants species at varying sequencing depths. We compared IndelEnsembler to INSurVeyor on the same datasets for the species B. Napus and Soybean (Fig. 5b). For both species, INSurVeyor is significantly more sensitive than IndelEnsembler. One issue noted by Liu et al. was that sensitivity for insertions was subpar when compared to deletions, and INSurVeyor fills this gap.

We called insertions using INSurVeyor on the 1047 A. Thaliana genomes studied by Liu et al., and we clustered them with the algorithm presented in ref. 32. The final callset consisted of 76,348 non-redundant insertions appearing in at least one *Arabidopsis thaliana* sample. The sizes of the insertions ranged from 50 bp to 15,343 bp (median 384 bp). Transposable elements are major components of plant genome and play an important role in creating structure variations. We identified 21,833 insertions (28.6%) as TE insertions, corresponding to three class I (LINE, Copia, and Gypsy retrotransposons) and eight class II (Helitron, En-Spm, Harbinger, hAT, Mariner, MuDR, Pogo, and Tc1 DNA transposons) superfamilies.

Most peaks in the distribution of insertion sizes (Fig. 6a) are contributed by TE insertions, with major peaks associated with events related to the Copia, Gypsy, Helitron, and hAT families of transposable elements (Supplementary Fig. 3a). Peaks at 123, 166, 292, and 444 bp mainly consisted of insertions of Copia retrotransposons. Peaks at 553 bp consisted of Copia and Helitron transposable elements. Lastly,

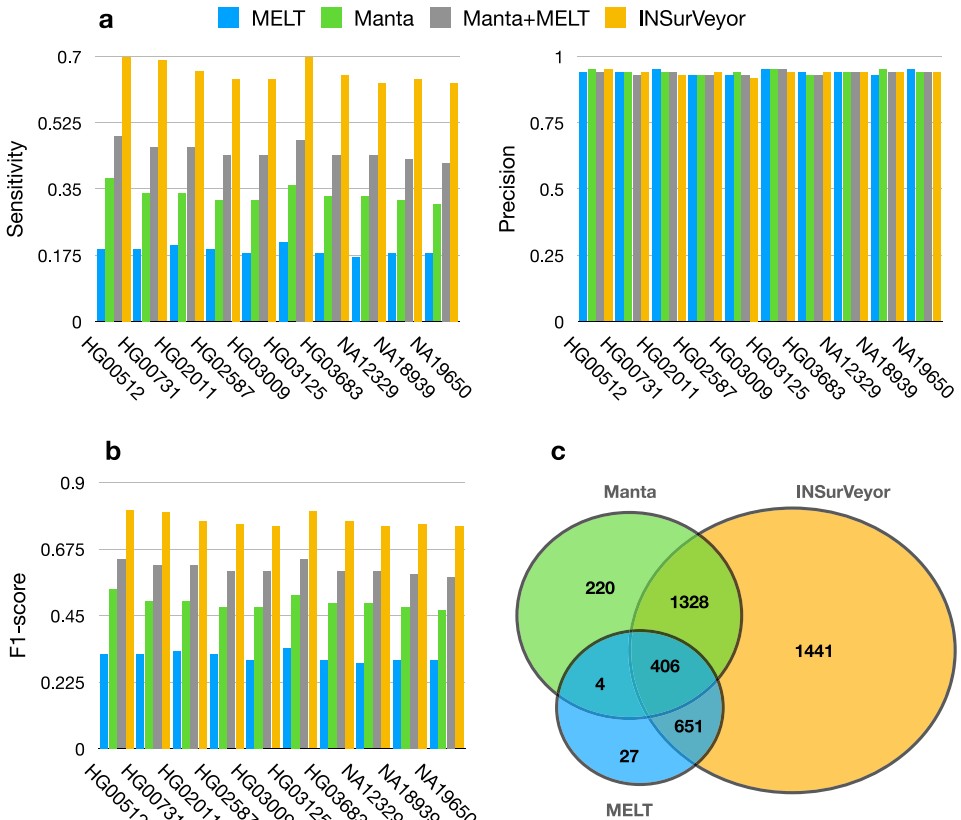

**Fig. 4 | Performance of the tested tools on the HGSVC2 benchmark. a** Sensitivity, precision and **b** F1-score of Manta, MELT, the union of the two and INSurVeyor (10 samples randomly picked are displayed here, a summary for the 34 samples is presented in Supplementary Fig. 8). Results are consistent with what was observed on HG002. **c** Venn diagram of the true positive calls per sample that are called by different combinations of the tools, averaged over the 34 genomes.

we found that 94% of the 759 bp insertion peak is consisted of hAT DNA transposons. Note that most of the peaks were either absent or barely noticeable in the original callset. The Helitron DNA transposons superfamily had the largest TE insertion event contribution (7216 insertions), followed by the Copia (6336 insertions) and MuDR (2872 insertions) retrotransposons families (Supplementary Fig. 3b). In total, we obtained 16,927 TE insertions occurring within 1.5 kb upstream (promoter) and downstream of the gene body. Specifically, we found 3879 genes overlapping with at least one TE insertion. We found that 28.86% of Copia, 27.31% of Gypsy retrotransposons insertions and 24.06% En-Spm DNA transposons insertions inserted into coding regions. These ratios were significantly higher than the others TE insertions (the corresponding ratios of LINE, MuDR, hAT, and Helitron were 9.26%, 4.21%, 4.55%, 1.87%, respectively, Fig. 6b). These findings indicate TE insertions extensively influence genes in *Arabidopsis thaliana*.

To explore the contribution of insertions to flowering time variation, we used our insertions to perform a genome-wide association study for flowering time of Spain 2008 and planting summer 2008[41]. We found a 982 bp insertion on Chr3, inserting into the fifth exon of *AT3G27570*, significantly influencing flowering time of both Spain 2008 and planting summer 2008 (Fig. 6c, d). *AT3G27570* encodes the sucrase, sucrase catalyzes the hydrolysis of sucrose to glucose and fructose. There has been a certain amount of evidence suggesting that sucrose promotes flowering[42,43]. King[44] reported that sucrose may regulate flowering by up-regulation of Flowering Locus T (FT, *AT1G65480*) expression. The 12 accessions with this insertion flowered later than those without it (Fig. 6c, d). This indicated that the 982 bp insertion is an important genetic variants for studying *Arabidopsis thaliana* flowering time. We also detected a 4428 bp insertion located on Chr5, inserting into the second exon of *AT5G21110* (Supplementary Fig. 3g), that

significantly associated with days until first open flower (DTF3) (Supplementary Fig. 3c), flowering time (under 10 °C and 16 °C) (Supplementary Fig. 3d, e) and rosette leaf number (RL) (Supplementary Fig. 3f)[45]. The accessions with this insertion flowered later, the days of first open flower delayed and the rosette leaf number increased compared to those without it. We also found a 445 bp insertion that significant associated with flowering time under 10 °C and was located on the second exon of *AT1G26570* (Supplementary Fig. 3h). *AT1G26570* is a UDP-glucose dehydrogenase (UGDH), which catalyzes the conversion of UDP-glucose to UDP-glucuronic acid[46]. The 11 accessions with this insertion flowered later than those without it (Supplementary Fig. 3i). Those examples showed that the insertions provided an important resource to find the causal phenotypic variation in *Arabidopsis thaliana*.

**An insertion catalogue for 3202 genomes in the 1000 genomes project**

INSurVeyor was used to generate a catalogue of insertions for the 3202 genomes sequenced by the New York Genome Center (NYGC)[36]. Note that, due to its speed, INSurVeyor was able to call insertions on 3202 30x human genomes in <3 days using a modest amount of resources (a cluster of 64 AWS r5.2xlarge 8-cores machines). Using a combination of the existing tools would have produced a less complete catalogue and would have required considerably higher computational resources.

After calling insertions for each individual genome, results containing long homopolymer runs of Cs or Gs were discarded. This is because the datasets were sequenced using NovaSeq 6000, and poly-Gs runs are most likely sequencing artefacts[47]. We found insertions into centromeres to have lower precision than insertions into other regions, so they were also removed. The remaining insertions were left-aligned and clustered across the samples so that within each cluster no two insertions would be >200 bp away, using the clustering algorithm

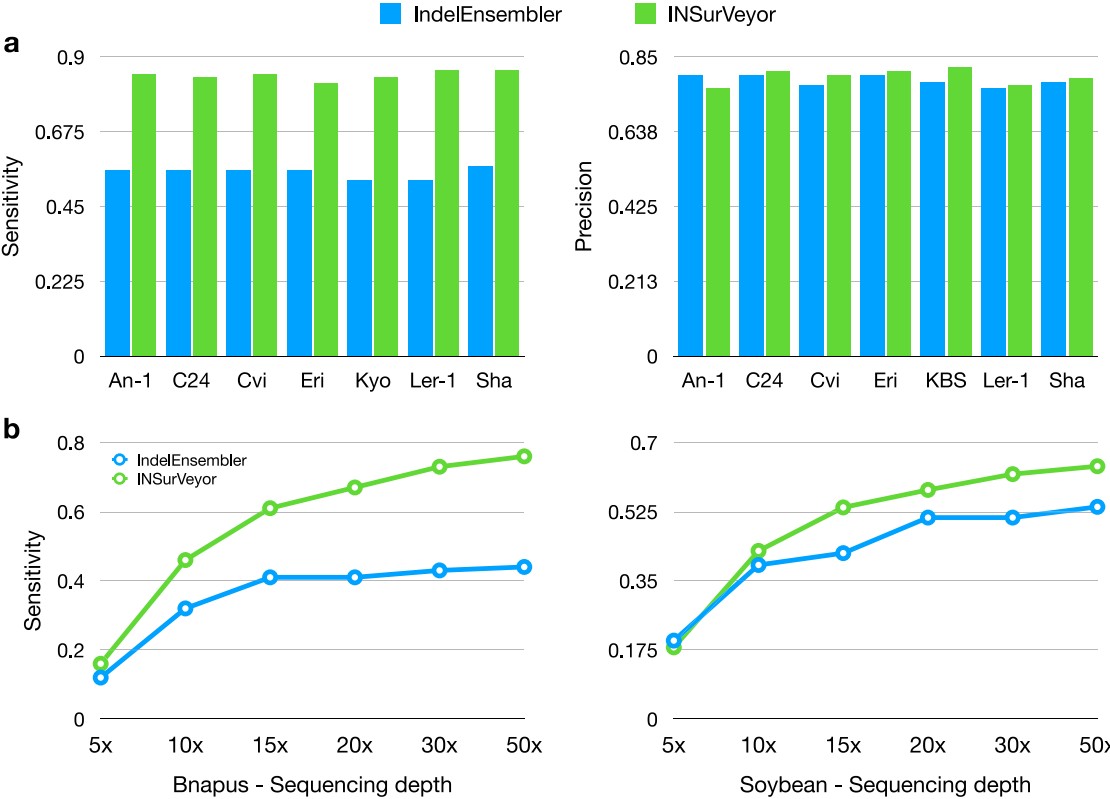

**Fig. 5 | Performance of IndelEnsembler and INSurVeyor on different plant samples. a** Performance of IndelEnsembler and INSurVeyor on predicting insertions on seven *Arabidopsis Thaliana* genomes. INSurVeyor is more sensitive. Furthermore, it is more precise in all samples except one. **b** We compared the sensitivity of IndelEnsembler to INSurVeyor on two more species of plants, B. Napus and Soybean, for different sequencing depths. In both species, INSurVeyor shows major improvements.

presented in ref. 32. Each cluster is a unique insertion in the population, and it is represented by the most common insertion in the cluster.

The final catalogue was composed of 148,934 insertions. Rare insertions (identified in <1% of the population) accounted for 83.9% of the catalogue, and 39.6% were singleton (i.e., private to one individual). Africans had more insertions per individual (median count 5119) than non-Africans (median count 4406) when compared to hg38. The number of insertions was similar between the remaining super-populations (Fig. 7a). The size distribution shows three peaks for mobile elements (ALU, SVA and LINE, Fig. 7b, also identified in ref. 13), and PCA analysis clearly segregates the superpopulations (Fig. 7c) and even subpopulations (Supplementary Fig. 10).

Along with the sequenced data, NYGC also produced a comprehensive catalogue of the SVs in 1000g using state-of-the-art tools for SV calling in a population[36]. We refer to it as 1000g-SV. Over 91% of the insertions in 1000g-SV were present in our catalogue. Only 4353 insertions were private to 1000g-SV, compared to 94,988 insertions private to INSurVeyor. When restricting to samples in HGSVC2, the validation rate of the 1000g-SV private SVs was 50%, compared to 80% for INSurVeyor. As expected, the validation rate for shared events was extremely high (99%). When considering the performance sample by sample, INSurVeyor was both more precise and more sensitive than 1000g-SV (Fig. 7e).

### STR expansions in ALU are potentially pathogenic and missed by existing methods

We further investigated the enrichment of novel calls by INSurVeyor in different regions of the genome compared to existing 1000g-SV calls (Fig. 8a), stratified by repeat content of the insertion site. For every repeat type, the number of INSurVeyor-private events is higher than the number of calls shared with 1000g-SV. SINE (6006 shared events,

17,133 private, 3.9-fold enrichment) and low complexity regions (4636 shared, 23,065 private, 6-fold enrichment) show the most enrichment in absolute terms. Low complexity regions are extremely difficult to resolve using short reads, as confirmed by Supplementary Fig. 2, so we focus our attention on insertions in SINE regions.

We partitioned insertions in SINE regions based on the repeat content of their inserted sequences. Unsurprisingly[48], SINE insertion was the major type. Interestingly, insertions of low-complexity sequences were the second most common. We identified 747 insertions of low-complexity sequences into reference SINEs in HGSVC2. Our callset contained 73% of them, while 1000g-SV contained 20% of them. Only 3 HGSVC2-supported calls were uniquely present in 1000g-SV, compared to 401 uniquely detected by INSurVeyor (Fig. 8c). Overall, the INSurVeyor callset contained 1655 low complexity sequences inserted into SINE regions. Upon closer inspection, the vast majority (potentially all) could be classified as short tandem repeat (STR) expansions, 74% of them located in the 3' tail of an Alu (Fig. 8d). Pathogenic STR expansions of Alus in intronic regions are known to cause neurodegenerative disorders such as CANVAS[7], different types of Spinocerebellar Ataxias[8–10] and Friedreich's Ataxia[11]. Our catalogue contained 562 intronic Alu STR expansions, and an additional 8 and 17 were in 3' UTRs and promoters, respectively. None were detected in 5' UTRs and coding regions (Fig. 8e). Among them, we could identify varying degree of polymorphism in *RFC1*, associated with CANVAS; *DAB1*, where we identified a known expansion of the motif AAAAT but not the pathogenic ATTTC, which is associated with SCA37; *ATXN10*, where we detected an expansion of the motif ATTCT in 12 individuals, all from Central and South America (where SCA10 is predominantly found), all in the intermediate range (30-799 copies); *FXN*, where we found short expansions of the GAA motif in 59 individuals (long expansions, usually > 2kb, are known to cause Friedreich's Ataxia). See

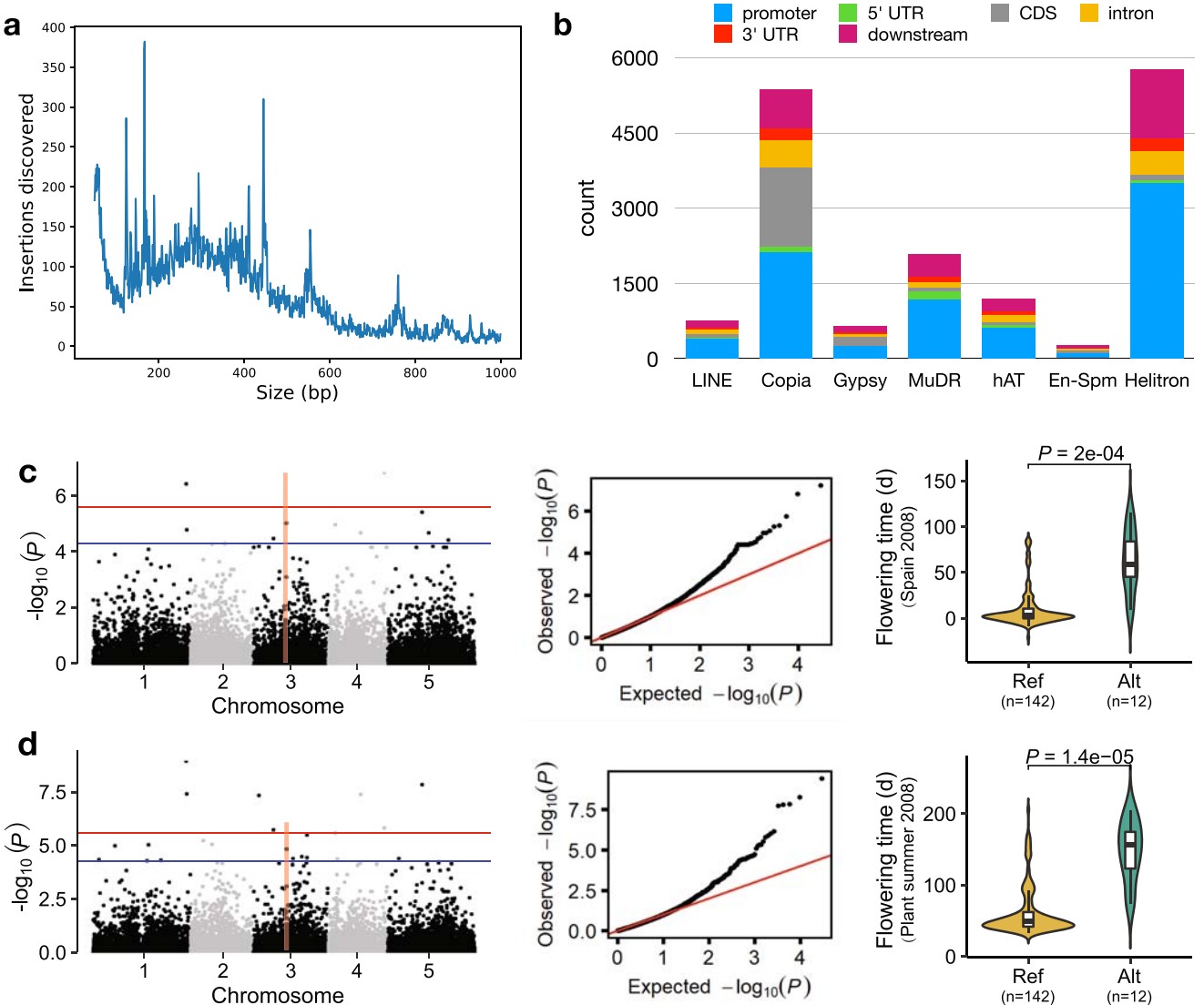

**Fig. 6 | Properties of the catalogue of insertions called in 1047 samples from the 1001 Genomes Project. a** Size distribution of the insertions discovered in 1047 samples of *Arabidopsis Thaliana*. The most prominent peaks are caused by insertions of transposable elements. **b** The counts of different TE insertions in genic regions. Genic regions include 1.5 kb upstream of the gene body. **c**, **d** A significant loci for flowering time of Spain 2008 (**c**) and summer 2008 (**d**). Left, Manhattan plots of insertions genome-wide association studies for flowering time of Spain 2008 (**c**) and plant summer 2008 (**d**). Blue and red horizontal lines indicate the significance thresholds of GWAS ($5.25 \cdot 10^{-5}$ and $2.63 \cdot 10^{-6}$, respectively). The

vertical line represents the candidate gene AT3G27570 on chromosome 3. Center, QQ plots for flowering time of Spain 2008 (**c**) and plant summer 2008 (**d**). Right, Violin plots showing the flowering time of accessions with different *AT3G27570* alleles for Spain 2008 (**c**) and plant summer 2008 (**d**) (*P*-values were determined using two-tailed Student's *t*-tests). Flowering time is significantly delayed in samples with the alternative allele compared to those with the reference allele. Boxplots in (**c**) and (**d**) show median (inner line) and inner quartiles (box). Whiskers extend to the highest and lowest values no greater than 1.5 times the inner quartile range.

Supplementary Table 1 for a list of loci of known pathogenic expansions in intronic Alus, along with the pathologies caused and the number of individuals in the 1000 Genomes Project with detected polymorphism. Only 107/562 (19%) of the intronic expansions were present in 1000g-SV. We ran ExpansionHunter Denovo[49], a specialised method for detecting STR expansions, and only 121/562 (22%) of them were detected. Our validation rate with HGSVC2 was 92%, which suggests that the expansions identified are reliable.

## Discussion

We presented a computational method, INSurVeyor, to identify insertions from paired-end WGS datasets. We tested several state-of-the-art callers, both specialised and general, on publicly available benchmark human genomes. INSurVeyor was more sensitive than all of them combined, and predicted a large number of true positives missed

by all other methods. It performed well across all types of insertions, and with the exception of insertions of low-complexity sequences, it predicted >85% of the insertions predicted by long reads. Given the exponential increase in datasets sequenced, speed and precision are of primary importance. INSurVeyor was both extremely precise and fast.

INSurVeyor is completely agnostic to the species analysed. Recently, Liu et al.[32] published IndelEnsembler, an ensemble SV caller, and used it to study a population of 1047 *Arabidopsis Thaliana* genomes. We called the insertions on benchmark datasets of different species of plants and showed that INSurVeyor has consistently higher sensitivity when compared to IndelEnsembler. We then generated a catalogue of insertions for the 1047 A. Thaliana used by Liu et al., and found multiple insertions that were previously missed that strongly correlate with phenotypes such as flowering time, days until open flower and rosette leaf number.

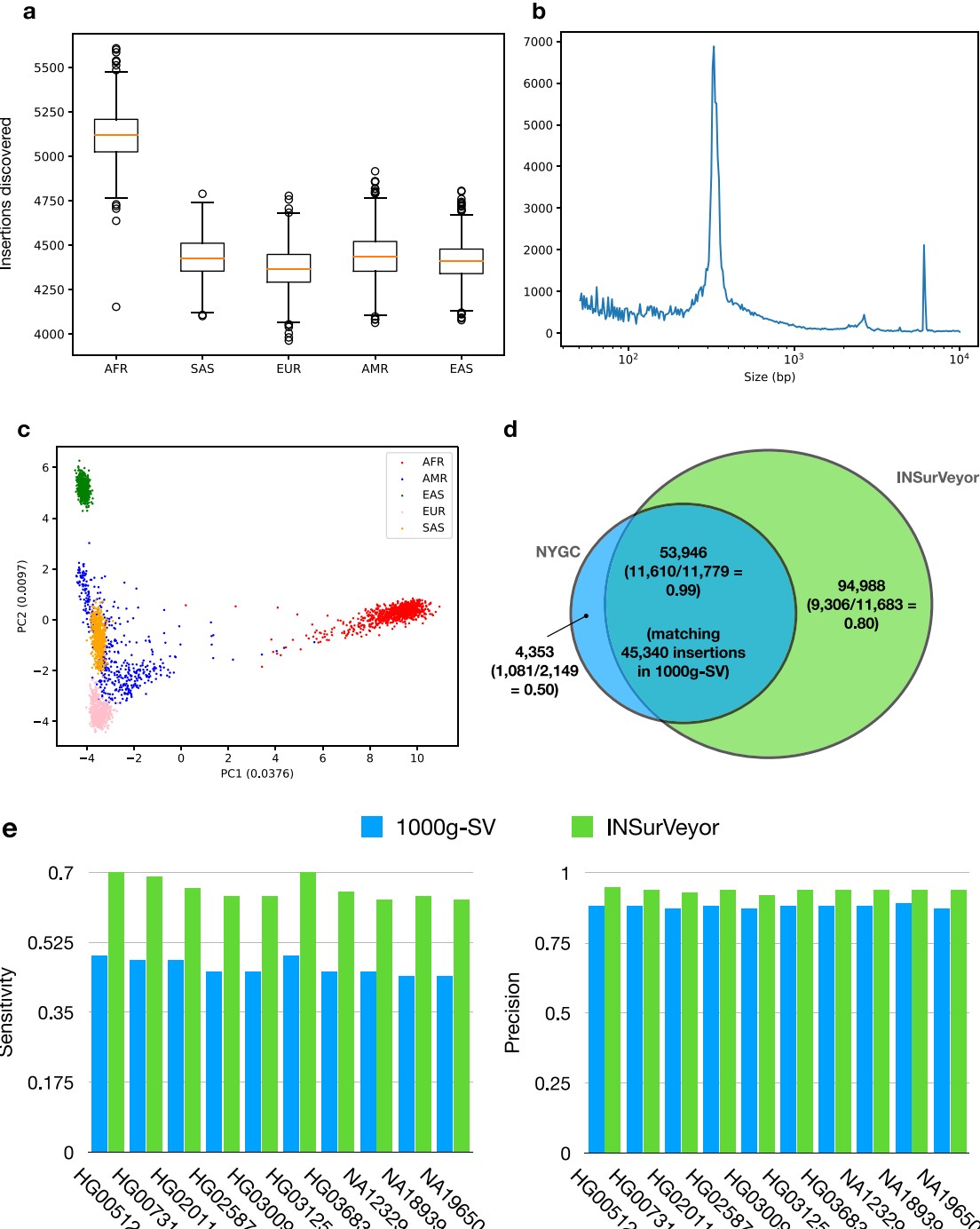

**Fig. 7 | Properties of the catalogue of insertions called in 3202 samples from the 1000 Genomes Project. a** Number of insertions called per superpopulation. Africans consistently have a higher number of insertions than other superpopulations when compared to hg38. The boxes contain values from the lower to the upper quartile, the line within the box is the median and the whiskers extend by 1.5 times the interquartile range. Circles represent data points outside of the whiskers. **b** Length distribution of the inserted sequences. The ALU, SVA and LINE peaks are all clearly present. **c** Principal component analysis (PCA) of the distribution of the insertions in the population clearly separates the superpopulations. **d** Number of private and shared calls between the 1000g-SV and the INSurVeyor callsets.

Between parentheses, the validation rates of calls in samples with long reads. Note that we match insertions as long as they are within 500 bp from each other, therefore a single insertion from 1000g-SV can match multiple insertions from INSurVeyor, and vice versa. For this reason, the number of 1000g-SV insertions with a match in INSurVeyor (45,340) is not the same as the number of INSurVeyor insertions with a match in 1000g-SV (53,946). Not only INSurVeyor has a large number of private events (94,988 compared to 4353 private to 1000g-SV), but also a much higher validation rate. **e** When evaluated sample by sample using HGSVC2, INSurVeyor is consistently more sensitive and precise (10 randomly picked samples shown here, a summary for the 34 samples is shown in Supplementary Fig. 9).

Finally, we demonstrated the advantages of having a fast, sensitive and precise caller by generating a catalogue of insertions for the 1000 Genomes Project samples resequenced by the New York Genome Center. Due to its speed, we were able to call the insertions for all 3202 samples in <3 days using modest resources. NYGC has generated a catalogue of SVs for the same samples by jointly using GATK-SV and svtools, the pipelines behind major studies, as well as in-house tools. Our final catalogue contained nearly three times as many insertions,

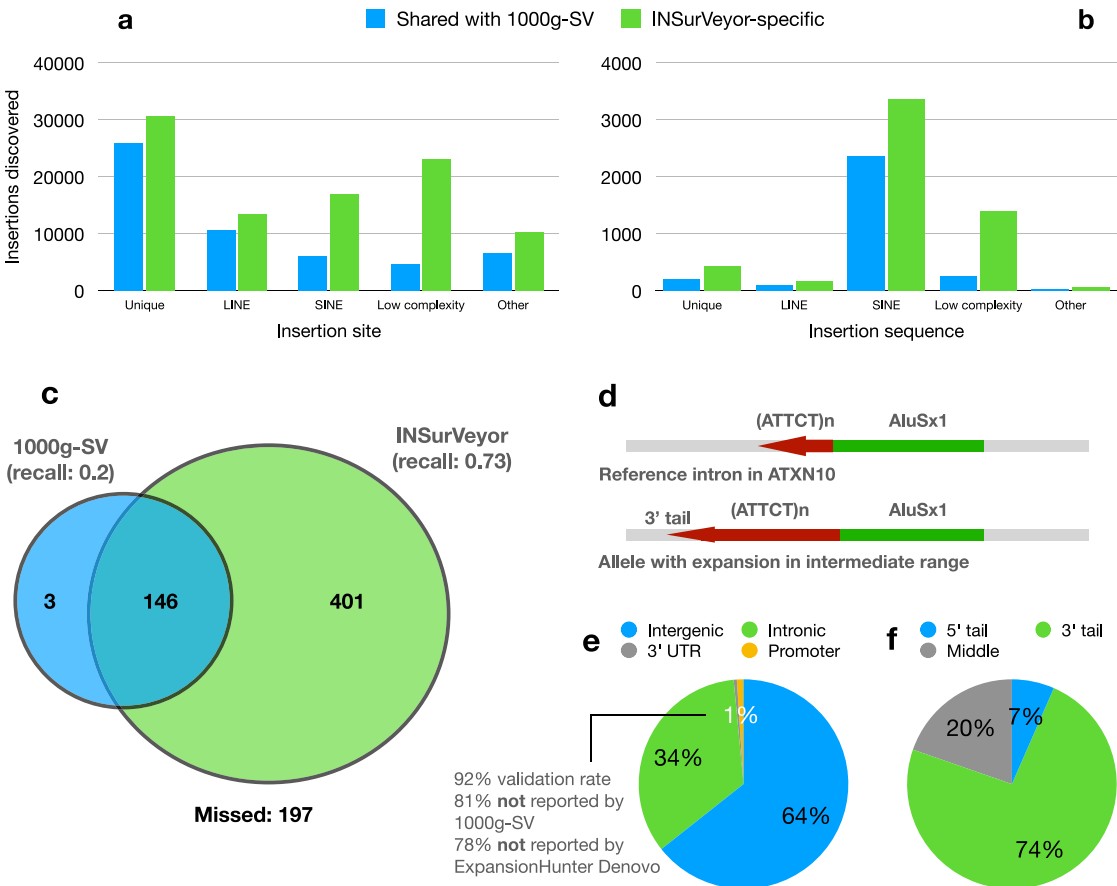

**Fig. 8 | Analysis of enriched regions and insertion types in the INSurVeyor dataset. a** Compared to the 1000g-SV dataset, INSurVeyor shows the most enrichment in SINE and low complexity regions (as annotated by RepeatMasker). **b** We classify insertions in SINE regions by repeat content of the inserted sequence. Most insertions into reference SINEs are by other SINE sequences. The second most frequent category is the insertion of low-complexity sequences, and they are mostly specific to INSurVeyor. **c** We identify 747 HGSVC2 calls as insertions of a low complexity sequence into a SINE. Our catalogue contains 73% of them, while only 20% are present in the 1000g-SV dataset. Only 3 are uniquely present in 1000g-SV and missed by INSurVeyor. **d** We observed that most (potentially all) insertions of low complexity sequences into SINE regions are due to STR expansions. One notable example is the expansion of the 3' tail of an AluSx1 element in an intron of the *ATXN10* gene. Very large expansions (≥800 copies) of the ATTCT motif result in SCA10. **e** 34% of the SINE STR expansions are in intronic regions, and most of them are not reported by 1000g-SV nor ExpansionHunter Denovo, a specialised tool. The validation rate when compared to HGSVC2 is 92%, which suggest most detected expansions are true positives. Some intronic ALU STR expansions are known to cause neurodegenerative diseases. **f** Most expansions (74%) happen in the 3' tail of an ALU element. We consider the 3'-most 30 bp of an ALU to be its 3' tail.

and 94,337 insertions were unique to it, while only 4387 insertions were unique to the NYGC catalogue. Using samples with benchmarks, the validation rate of INSurVeyor-specific calls was far superior (80%) to that of NYGC-specific calls (50%).

Our catalogue was enriched over the published one across all types of repeats, and we expect that INSurVeyor will provide biologists and bioinformaticians with a powerful tool to uncover a degree of variation that was previously missed. For example, a handful of short tandem repeat expansions in reference Alu elements in intronic regions are known to cause neurodegenerative pathologies. INSur-Veyor detected polymorphism consistent with the literature in all of them. Furthermore, it reported 562 expansions in intronic Alu elements across the human genome, most of them missed even by specialised tools, and with high validation rates. We hypothesise that previously missed clinically relevant loci may be among them.

One type of insertion where long reads still have a clear advantage is when the inserted sequence is a low complexity sequence. This is due to the repetitive nature of those sequences, as well as other technical reasons, which make mapping the reads sequenced from them a challenging task. Although we believe short reads may have intrinsic limitations in detecting such events, we also believe that there may still be room for improvement and efforts should be put into

reducing the gap with long reads, especially with the many exciting large-scale sequencing efforts currently underway.

## Methods
### Excluding tandem duplications
In both SV catalogues that we use as benchmark (GIAB-SV and HGSVC2), part of the insertions are due to tandem duplications. However, short-read based SV callers such as Manta and Delly report tandem duplications as a separate class of SVs from insertions. Furthermore, tandem duplications are outside of the scope of INSurVeyor, MELT, xTea and Pamir.

Let $R[1..n]$ be a region of the reference, and $R[0]$ and $R[n+1]$ be the base pair immediately before and immediately after it, respectively. Let $I$ be an insertion that inserts a sequence $S$ between $R[i]$ and $R[i+1]$, $i \in [0..n]$. $I$ is a tandem duplication if $S = R[i+1..n]R[1..i]$ ($R[1..0]$ and $R[n+1..n]$ are defined as empty strings). This is because the resulting sequence is $R[1..i]SR[i+1..n] = R[1..i]R[i+1..n]R[1..i]$ $R[i+1..n] = RR$. Therefore, we employ the following procedure to decide whether a benchmark insertion $I$ is a tandem duplication and should be excluded.

- Let $S$ be the inserted sequence of $I$. A base sequence $B$ of $S$ is a sequence such that $S$ can be obtained by concatenating $B \alpha$ times

(where $\alpha \geq 1$). We use TRF[50] to find the shortest base sequence of $S$ that is at least 50 bp. We call it $B'$.

- Let $l$ be the length of $B'$, and $p$ be the genomic coordinate of the insertion site on the reference. For every $n \in \{0 .. l\}$, we divide $B'$ into $P_n$ and $S_{l-n}$, where $P_n$ is the first $n$ characters of $B'$ and $S_{l-n}$ is the last $l-n$ characters of $B'$. We perform a Smith-Waterman alignment (match = 1, mismatch = -4, gap opening = -6) between the genomic region $[p-(l-n),p]$ and $S_{l-n}$, and between the genomic region $[p, p+n]$ and $P_n$. If at least 80% of $P_n$ and $S_{l-n}$ are covered by the alignment, the insertion is marked as a duplication. Supplementary Fig. 8 illustrates this with an example and explains the rationale behind the algorithm.

### Tested software

For comparison, we aimed at selecting the best two representative from three distinct categories: general SV callers, mobile element insertion (MEI) callers and novel insertion callers. For objectiveness of the evaluation, we require the tool to explicitly call the insertions, without a need for the user to interpret or transform the results. All software were run with default parameters, and insertions of at least 50 bp were retained.

For SV callers, we relied on a recent in-depth evaluation of the 10 most popular SV callers[15], which shows that Manta and GRIDSS generally perform the best, followed by Delly and Lumpy. Out of the four, only Manta and Delly explicitly report insertions, so we selected them.

A review of specialised MEI callers[26] found that MELT, Mobster and Retroseq had the best performance out of the tested tools. We have previously found[29] that among the three, MELT performed the best. MELT is widely adopted and it is used by several projects such as the 1000 Genomes Project and gnomAD-SV. We also included a very recent tool, xTea, which was shown to perform better than MELT on the datasets tested by the authors.

Finally, we tested PopIns2 and Pamir, two recent novel insertion callers. However, PopIns2 provided a very low number of insertions on the datasets we tested, so we left it out of the comparison.

All software was run on a server equipped with two Intel Xeon 6252 and 1.5 TB of RAM, running Linux Ubuntu 20.04. Multithreading was limited to 8 cores.

### Comparing insertions

An insertion is defined by an insertion site and an inserted sequence. Because xTea and MELT do not report the actual inserted sequence, in order to have a fair comparison between the tools, we use a *relaxed* comparison. Given two insertions $i_1$ and $i_2$, we say that $i_1$ and $i_2$ match if their insertion sites are within 500 bp of each other.

Only when explicitly mentioned and when all the tools report the inserted sequences, we use a *strict* criterion. When the inserted sequence is very long (typically much longer than insert size), Manta and INSurVeyor may only report a prefix and a suffix of the inserted sequence. When using the strict criterion, we discard such insertions. Given two insertions $i_1$ and $i_2$, let $S_1$ and $S_2$ be their inserted sequences, and $l_1$ and $l_2$ be the lengths of $S_1$ and $S_2$, respectively. $i_1$ and $i_2$ match if their insertions sites are within 500 bp, (b) $|l_1 - l_2| \leq 500$ and (c) if the Smith-Waterman alignment score between $S_1$ and $S_2$ is greater or equal than $min(l_1, l_2)$ (match = 2, mismatch = -2, gap opening = -4).

### Determining the repeat content in inserted sequences and insertion sites of human samples

In order to determine the repeat content of the inserted sequences, we used RepeatMasker. In particular, we classified as SINE sequences that were fully (50 bp tolerance) annotated as SINE by RepeatMasker. Similarly for LINE. For low complexity, we accepted sequences fully (50 bp tolerance) annotated as low complexity or as simple repeats. All the insertions that fit in neither categories are classified as "other".

In order to classify the insertion sites, we overlapped them with the RepeatMasker track provided by the UCSC Genome Browser.

### Identification of TE insertions in Arabidopsis thaliana

Whole-genome resequencing data of 1047 *Arabidopsis thaliana* accessions and paired-end alignments on the reference genome of Tair10 (Arabidopsis Col-0) were obtained from ref. 32. The TE library of *Arabidopsis thaliana* was downloaded from https://arabidopsis.org/download_files/Genes/TAIR10_genome_release/, including the LINE, Copia, Gypsy LTR retrotransposons and eight other DNA transposons (Helitron, En-Spm, Harbinger, hAT, Mariner, MuDR, Pogo, and Tc1). After that, sequences of all insertions were mapped to the TE library using blastn (v.2.6.0) based on the "80-80 rule"[51]. If both the identity and coverage of each insertion reached 80%, then the insertion was defined as a TE insertion. Using this strategy, we extracted TE insertions in the 1047 *Arabidopsis thaliana* genomes.

### GWAS using the Arabidopsis thaliana insertions dataset

The phenotypes of flowering time under 10 °C and 16 °C were obtained from The 1001 Genomes Consortium[52]. Eight flowering related phenotypes and thirty-four flowering time phenotypes in simulated seasons were download from the Arapheno database[41,45]. Insertions with a minor allele frequency (MAF) >0.01 were used for genome-wide association study (GWAS). GWAS was performed for the all traits with GEMMA (v0.98.1)[53]. The population structure was generated as covariates using admixture (v1.3.0), as well as an IBS kinship matrix derived from SNP and small InDels calculated by GEMMA. The cutoff for determining significant associations was set as -log10(1/n), where n represents the total number of insertions.

### Reporting summary

Further information on research design is available in the Nature Portfolio Reporting Summary linked to this article.

## Data availability

The insertions data generated in this study have been deposited in EBI. The insertions calls from 3202 samples from the 1000 Genomes Project are under project PRJEB59423 [https://www.ebi.ac.uk/ena/browser/view/PRJEB59423]. The clustered calls are available under analysis ERZ16007666 [https://www.ebi.ac.uk/ena/browser/view/ERZ16007666], while the single-sample calls are available under analysis ERZ16007665 [https://www.ebi.ac.uk/ena/browser/view/ERZ16007665]. The insertion calls from 1047 *Arabidopsis Thaliana* from the 10001 Genomes Project are under project PRJEB58052 [https://www.ebi.ac.uk/ena/browser/view/PRJEB58052]. The clustered calls are available under analysis ERZ14864777 [https://www.ebi.ac.uk/ena/browser/view/ERZ14864777], while the single-sample calls are available under analysis ERZ16031661 [https://www.ebi.ac.uk/ena/browser/view/ERZ16031661]. Sequencing data for HG002 was downloaded from NCBI (accessions SRR1766442 to SRR1766486). PacBio HiFi data for HG002 was downloaded from NCBI under accession code [PRJNA586863]. The HG002 benchmark catalogue and the list of tier 1 regions were downloaded from https://ftp-trace.ncbi.nlm.nih.gov/ReferenceSamples/giab/data/AshkenazimTrio/analysis/NIST_SVs_Integration_v0.6/. Information on accessing the 3202 CRAM files for the 1KGP project produced by NYGC can be found at https://www.internationalgenome.org/data-portal/data-collection/30x-grch38. The Phase 2 benchmark calls produced by HGSVC are available at https://www.internationalgenome.org/data-portal/data-collection/hgsvc2. The SV catalogue produced by NYGC was downloaded from http://ftp.1000genomes.ebi.ac.uk/vol1/ftp/data_collections/1000G_2504_high_coverage/working/20210124.SV_Illumina_Integration/1KGP_3202.gatksv_svtools_novelins.freeze_V3.wAF.vcf.gz. RepeatMasker annotations for hg19 and hg38 were downloaded from the UCSC Table Browser. Data from the 1001 Genomes Project was downloaded from

NCBI under accession code [PRJNA273563]. The TE library of *Arabidopsis Thaliana* was downloaded from https://arabidopsis.org, under Download/Genes/TAIR10 genome release/TAIR10 transposable elements. Phenotypic data was downloaded from the Arapheno database.

## Code availability

Source code for INSurVeyor is available for download at https://github.com/kensung-lab/INSurVeyor.

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

## Acknowledgements

This work was supported by the Hong Kong Genome Institute.

## Author contributions

R.R. developed and implemented the method, performed the benchmarking and conducted the analysis of the human dataset, under the guidance of W.-K.S. D.-X.L. conducted the analysis of the Arabidopsis Thaliana dataset, under the guidance of Q.-Y.Y. C.H.A., Y.-T.C. and A.Y.T.L. performed additional quality check by manually inspecting and commenting on a large number of calls, and helped with the analysis of the human dataset. R.R., D.-X.L. and W.-K.S. wrote the manuscript.

## Competing interests

The authors declare no competing interests.
