## [Peer Review File · Nature Communications]

INSurVeyor: improving insertion calling from short read sequencing dataREVIEWER COMMENTS

Reviewer #1 (Remarks to the Author):

The authors present INSURVeyor, a tool for calling insertions from short reads sequencing data. The current software ecosystem has plenty of headroom for optimizations, especially for insertions, and it appears that INSURVeyor may fill this gap. The manuscript is well written and provides a promising overview of the capabilities of INSURVeyor. Please find my comments below.

Upon installation (on a rather fresh Ubuntu 22.04 server), I ran into errors with building htlib. I noticed that the version packaged with INSURVeyor is also not the most recent (v1.13, July 2021). After installing htlib with conda the cmake command resulted in compiler issues where the C compiler and CXX compiler were unknown. Setting the CC and C++ environmental variable did not change the error message. Manually editing CMakeCache.txt fixed this issue, but the installation ran into another issue later:

CMake Error: The following variables are used in this project, but they are set to NOTFOUND.

Please set them or make sure they are set and tested correctly in the CMake files:

HTS_LIB

linked by target "normalise" in directory /home/wdecoster/repositories/INSURVeyor

linked by target "filter" in directory /home/wdecoster/repositories/INSURVeyor

linked by target "dc_remapper" in directory /home/wdecoster/repositories/INSURVeyor

linked by target "call_insertions" in directory /home/wdecoster/repositories/INSURVeyor

linked by target "clip_consensus_builder" in directory /home/wdecoster/repositories/INSURVeyor

linked by target "add_filtering_info" in directory /home/wdecoster/repositories/INSURVeyor

linked by target "reads_categorizer" in directory /home/wdecoster/repositories/INSURVeyor

-- Generating done

CMake Generate step failed. Build files cannot be regenerated correctly.

I again attempted to run build_htlib.sh, but then zlib.h was missing.

At this point I think I did more to troubleshoot those installation issues than what you can expect from (novice) users, and further facilitating the installation of your tool is up to the developers. Providing a singularity container is appreciated, but not a sufficient solution for most users. I would like to urge the authors to add their tool to bioconda, which is the de facto expected source of bioinformatics software and further makes it easy to contain software in reproducible environments, as well as use your tool easily through workflow managers such as snakemake and nextflow.

The upset plot in Figure 2b appears to show that not a single insertion is supported by all 6 variant callers - which I find highly surprising. Is there truly no overlap between calls supported by INSURVEYOR and the other callers? I assume there has to be an overlap, and this figure is erroneously showing the contrary or constructed in a misleading way. I assume the authors misinterpreted how to construct an upset plot.

It is not entirely clear how insertions were compared between the callers. Supplementary note 3.6 outlines the parameters used, but there is no reference to a tool or code on how to do this. Note that the commonly accepted method for comparing structural variant call sets is Truvari (<https://github.com/ACEnglish/truvari>). Can the authors confirm that they obtain similar results with their tool?

I do not fully understand the choice for SVIM as a structural variant caller for long reads to compare with, and the provided reference does not show that it is the most accurate SV caller. While SVIM is a great tool, recommended would be using Sniffles2. I however do not think this would majorly impact the conclusions presented in this manuscript, but it may warrant some nuance when comparing short read with long read callers.

While the order of the bars is helpful, it is rather confusing that in figure 3 the color for Delly and SVIM are highly similar. The Venn diagrams also use another color for INSURVEYOR than the bar charts.

Can the authors comment on why they removed long homopolymer runs of Cs or Gs from the 1000 genomes call set? Is this because the NovaSeq uses 2-color chemistry and the absence of signal/nucleotide at the end of short fragments is the same as the G nucleotide?

I can imagine that it is beneficial for users to get a highly complete set of variants, without limiting to insertions. Can the authors provide recommendations with which tool to combine INSURVEYOR, possibly Manta, to get a highly complete set of all types of SVs?

Reviewer #2 (Remarks to the Author):

INSurVeyor presents a method that will be very welcome to the community and the results are compelling.

Some work needs to be done on the software to make it easier to install. While we were able to get the tool running, it required more effort than I think the average user will be willing to expend.

The major issue that I have is that there is that the method will be very difficult to use on a population, which I expect will be a primary use case. INSurVeyor, quickly runs on individual samples, creating a well-formatted VCF. As far as we can, INSurVeyor does not provide a way to merge those VCFs into a single set of population calls with genotypes for all samples for all unique variants. I don't think this should prevent its publication, but it will have a major effect on its impact.

Reviewer #3 (Remarks to the Author):

The paper "INSurVeyor: improving insertion calling from short reads sequencing data," by Rajaby et al., suggests that INSurVeyor is a tool that utilizes short reads to identify insertions only precisely compared to other methods. They supported a catalog of insertions for 1,047 *Arabidopsis thaliana* genomes from the 1001 Genomes Project and 3,202 human genomes from the 1000 Genomes Project. INSurVeyor identifies insertions with higher accuracy, suggesting it to be a complementary tool that researchers can use in pipelines with other SV callers using short reads.

Introduction:

In the section "Nowadays, insertions are usually detected using either long read sequencing technologies (like PacBio HiFi reads) or paired-end short read

sequencing technologies (like Illumina)." Please, add also ONT as it is long-read technology, and it is capable of detecting insertions as well.

"However, not all SV callers explicitly report insertions, and when they do the sensitivity is low." Please, add a reference.

E.g., <https://doi.org/10.1038/s41576-019-0180-9>, <https://doi.org/10.1186/s13059-019-1828-7>, or any paper.

Because you already compared INSURVEYOR to a combination of callers. In the section where you give examples of short-read SV callers, it is worth mentioning MetaSV and Parliament.

In Figure 1, please, explain what the b, c, and d graph means.

Results:

What is the limitation for SVs calling length-wise?

“We downloaded a 50x PacBio HiFi dataset for HG002 (PRJNA586863) and ran SVIM [23], the most accurate SV caller for PacBio reads according to a recent benchmark”. I suggest using Sniffles2, which shows more accurate results.

“We partitioned the insertions into three major categories, according to their inserted sequences: mobile elements (1,282, 32%), low complexity sequences (1,111, 27.7%) and others (1,615, 40.3%)”. Could the authors discuss how they split them?

Minor comments.

The authors use comma-separated numbers, but in some cases, they do not.

We thank the reviewers for the helpful comments. We noted that the reviewers encountered difficulties in installing the software. We have tried to simplify the installation process as follows:

- Clearer documentation on the requirement for building htlib (autoconf, zlib and how to obtain them on ubuntu)
- Removed some of the pre-requirements for build_htlib.sh (bz2, lzma, libcurl)
 - a separate build_htlib_full.sh is provided that include supports for bz2, lzma and libcurl is also provided, in case it is needed
- Building the included htlib is recommended. However, if for any reason this did not happen, INSURVeyor will search for htlib installed in the system (>=1.13 is required)

We expect to be able to further improve the process as the software is used by a variety of users and we receive more feedback.

Reviewer 1

At this point I think I did more to troubleshoot those installation issues than what you can expect from (novice) users, and further facilitating the installation of your tool is up to the developers. Providing a singularity container is appreciated, but not a sufficient solution for most users. I would like to urge the authors to add their tool to bioconda, which is the de facto expected source of bioinformatics software and further makes it easy to contain software in reproducible environments, as well as use your tool easily through workflow managers such as snakemake and nextflow.

We have taken steps to improve the installation process, outlined at the beginning of this letter. We agree that having a conda package is a good idea, and we submitted one for review to conda-forge. We will update the README once the package is made public.

The upset plot in Figure 2b appears to show that not a single insertion is supported by all 6 variant callers - which I find highly surprising. Is there truly no overlap between calls supported by INSURVeyor and the other callers? I assume there has to be an overlap, and this figure is erroneously showing the contrary or constructed in a misleading way. I assume the authors misinterpreted how to construct an upset plot.

Sorry for the unclear description. The upset plot aims at comparing the number of true positives for INSURVeyor alone vs using combinations of different existing tools, therefore entries for INSURVeyor + other tools are excluded. Indeed this was not clear from the caption of Fig. 2b. We changed it to: "(b) The number of predicted TPs and the running time in minutes for INSURVeyor and for different combinations of existing tools (sorted by number of TPs, top 20 showed)."

It is not entirely clear how insertions were compared between the callers. Supplementary note 3.6 outlines the parameters used, but there is no reference to a tool or code on how to do this. Note that the commonly accepted method for comparing structural variant call sets is Truvari (<https://github.com/ACEnglish/truvari>). Can the authors confirm that they obtain similar results with their tool?

We used a small and simple in-house program. We published it in <https://github.com/Mesh89/SVComparator>. We added a small README, but this version is not for the general public. We are developing a more usable, complete and customisable SV comparison software. The results are obtained with

```
compare-ins $BENCHMARK_VCF $CALLED_VCF /dev/null /dev/null 500 --report [--ignore-seq]
```

--ignore-seq is omitted when the sequences are compared.

We checked if Truvari provides similar results. Because Truvari was not immediately applicable to HG002 (it complains about contig lengths missing in the benchmark VCF), we tested it on HG00512 (one of the 1000g samples) with the flags "--multimatch --pctsim 0 --pctsize 0 -r 500" to simulate the comparison method in the paper. The results match for INSURVeyor (Fig. 4 in the manuscript):

```
"precision": 0.9511120898480511,  
"recall": 0.6955296769346356,
```

However, for Manta, the recall is lower than what we measured with our tool (we measured 0.38).

```
"precision": 0.9506762132060461,  
"recall": 0.29902329075882794,
```

While for MELT, we get no output, and the error message:

```
[E::vcf_format] Invalid BCF, the INFO index 33 is too large
```

When comparing inserted sequences, we were not able to test Truvari as it expects the inserted sequence in the ALT field, while we use the symbolic representation and we place the inserted sequence in the INFO/SVINSSEQ field.

Overall, it seems that using Truvari would penalise Manta. However, because we are not sure about the reason of the discrepancy, we prefer to retain the better Manta recall measured with our tool.

I do not fully understand the choice for SVIM as a structural variant caller for long reads to compare with, and the provided reference does not show that it is the most accurate SV caller. While SVIM is a great tool, recommended would be using Sniffles2. I however do not think this would majorly impact the conclusions presented in this manuscript, but it may warrant some nuance when comparing short read with long read callers.

In Figure 2 of “Nicolas Dierckxsens, Tong Li, Joris R Vermeesch, and Zhi Xie. A benchmark of structural variation detection by long reads through a realistic simulated model. Genome Biol, 22(1):342, 12 2021.”, SVIM shows the highest “TOTAL SCORE” for “PACBIO” (lower left quadrant). We also tried cuteSV, but found that the results were almost identical. On suggestion of the reviewer we tried Sniffles2, and it does provide marginally higher recall (~1%). This is reflected in a higher recall for MEI and Other, but lower for Low complexity regions (although the differences are in the order of 1-3%). Given the higher recall, we replaced SVIM with Sniffles2, and updated the manuscript with the new numbers and figures. In particular, Sniffles2 has an impressive 97% recall on LINE insertions compared to 86% for SVIM, so we removed the observation that INSURVEYOR has higher recall (91%) for LINE insertions.

While the order of the bars is helpful, it is rather confusing that in figure 3 the color for Delly and SVIM are highly similar. The Venn diagrams also use another color for INSURVEYOR than the bar charts.

We have changed the colour of Sniffles to a dark green, in both Fig. 3 and S2, that should hopefully be more distinguishable from the other six callers. We also changed the colours of the Venn diagram to reflect the colours of the bar chart.

Can the authors comment on why they removed long homopolymer runs of Cs or Gs from the 1000 genomes call set? Is this because the NovaSeq uses 2-color chemistry and the absence of signal/nucleotide at the end of short fragments is the same as the G nucleotide?

Yes. The NYGC 1000 genomes dataset (Byrka-Bishop et al 2022 Cell, PMID:36055201) is sequenced using Illumina NovaSeq 6000 that utilises 2-channel sequencing by synthesis technology. Base G is unlabeled (Stoler et al 2021 NAR Genom Bioinform, PMID:33817639) and poly-G artefacts appear when the dark base G is called after synthesis has terminated. Such artefacts are well recognised by Illumina and they made specific Poly-G trimming bioinformatics tool (https://support.illumina.com/content/dam/illumina-support/help/Illumina_DRAGEN_Bio_IT_Platform_v3_7_1000000141465/Content/SW/Informatics/Dragen/PolyG_Trimming_fDG.htm). We noted that (1) there is no Poly-G trimming as part of NYGC processing pipeline and (2) Poly-G artifacts will be assigned high quality values in FASTQ sequences. Furthermore, we found that most poly-G and poly-C could not be validated by the benchmark datasets. Therefore, we removed insertions with long homopolymer runs of Cs and Gs.

The number of removed insertions per genome is very low (7.4 on average), but over thousands of genomes the number becomes more relevant. We added a comment in the manuscript: “This is because the datasets were sequenced using NovaSeq 6000, and poly-Gs runs are most likely sequencing artefacts [46]”.

I can imagine that it is beneficial for users to get a highly complete set of variants, without limiting to insertions. Can the authors provide recommendations with which tool to combine INSURVEYOR, possibly Manta, to get a highly complete set of all types of SVs?

Yes, we recommend Manta as it is an extremely good SV caller, in our experience. We added a section in the github README called “Complete SV callset” that shows how to combine Manta and INSURVEYOR. The section may change as new tools are developed.

Reviewer 2

Some work needs to be done on the software to make it easier to install. While we were able to get the tool running, it required more effort than I think the average user will be willing to expend.

We have taken steps to improve the installation process, outlined at the beginning of this letter. Furthermore, we submitted the software to conda-forge, and we will add it to the README once it goes public.

We are certain that the feedback of the users will allow us to further improve in this regard.

The major issue that I have is that there is that the method will be very difficult to use on a population, which I expect will be a primary use case. INSURVEYOR quickly runs on individual samples, creating a well-formatted VCF. As far as we can, INSURVEYOR does not provide a way to merge those VCFs into a single set of population calls with genotypes for all samples for all unique variants. I don't think this should prevent its publication, but it will have major effect on its impact.

Currently the clustering software is at <https://github.com/Mesh89/SurVClusterer>

This is a more general (i.e., able to cluster different types of SVs and with more parameters) version of the clusterer used to generate the published dataset. However, it is still in active development, and requires more testing, documentation and optimisations.

Nonetheless, it can be used to cluster insertions from different samples. We added a section to the INSURVEYOR github README that points to it.

Alternatively, more mature software (<https://github.com/DecodeGenetics/svimmer> and <https://github.com/mkirsche/Jasmine> come to mind) should work, although we have not personally tried them.

Reviewer 3

In the section “Nowadays, insertions are usually detected using either long read sequencing technologies (like PacBio HiFi reads) or paired-end short read sequencing technologies (like Illumina)”. Please, add also ONT as it is long-read technology, and it is capable of detecting insertions as well.

Thanks for the comment. We included mention to ONT nanopore sequencing as a long read technology.

“However, not all SV callers explicitly report insertions, and when they do the sensitivity is low.” Please, add a reference.

E.g., <https://doi.org/10.1038/s41576-019-0180-9>, <https://doi.org/10.1186/s13059-019-1828-7>, or any paper.

We added a reference to the Mahmoud et al. suggested by the reviewer. We also qualified the subsequent statement “When applied to recent comprehensive benchmark catalogues of SVs, all the callers we tested consistently detected less than 40% of the insertions” by referring to Section 2.2.

Because you already compared INSURveyor to a combination of callers. In the section where you give examples of short-read SV callers, it is worth mentioning MetaSV and Parliament.

We added the following paragraph to the Introduction: “Furthermore, several meta-callers have been developed, such as MetaSV [34] and Parliament2 [49], which integrate the output of multiple callers in order to increase recall. However, as shown in Fig. 2, combining multiple callers results in a higher runtime and there is a sharp diminishing return for the increase in recall as more methods are added. Another drawback is an increased number of false positives [32].”

In Figure 1, please, explain what the b, c, and d graph means.

We added the following sentence to the caption of Figure 1: “This is achieved by three separate modules: the remapping module (b) aims at predicting transpositions; the local assembly module (c) aims at predicting novel insertions, while the consensus overlap module (d) predicts small insertions.”. The inner workings of each module are explained in details in the main text and in the supplementary.

What is the limitation for SVs calling length-wise?

There is no limitation to the length of the insertions called. However, for very long novel insertions (where de novo assembly is required), INSURveyor may not be able to build the full sequence, and only provide the prefix and the suffix of it, and mark the insertions with the flag INCOMPLETE_ASSEMBLY. We added a subsection “Incomplete long novel insertions” to the README where we explain this.

“We downloaded a 50x PacBio HiFi dataset for HG002 (PRJNA586863) and ran SVIM [23], the most accurate SV caller for PacBio reads according to a recent benchmark”. I suggest using Sniffles2, which shows more accurate results.

This observation was shared by Reviewer 1. We report the answer to the comment:

In Figure 2 of “Nicolas Dierckxsens, Tong Li, Joris R Vermeesch, and Zhi Xie. A benchmark of structural variation detection by long reads through a realistic simulated model. Genome Biol, 22(1):342, 12 2021.”, SVIM shows the highest TOTAL SCORE for PACBIO (lower left quadrant). We also tried cuteSV, but found that the results were almost identical.

On suggestion of the reviewer we tried Sniffles2, and it does provide marginally higher recall (~1%). This is reflected in a higher recall for MEI and Other, but lower for Low complexity regions (although the differences are in the order of 1-3%). Given the higher recall, we replaced SVIM with Sniffles2, and updated the manuscript with the new numbers and figures. In particular, Sniffles2 has an impressive 97% recall on LINE insertions compared to 86% for SVIM, so we removed the observation that INSURveyor has higher recall (91%) for LINE insertions.

“We partitioned the insertions into three major categories, according to their inserted sequences: mobile elements (1,282, 32%), low complexity sequences (1,111, 27.7%) and others (1,615, 40.3%)”. Could the authors discuss how they split them?

We added a reference to Supplementary Section 3.7, where we discuss the use of RepeatMasker for categorising insertion sequences:

We partitioned the insertions into three major categories, according to their inserted sequences: mobile elements (SINE and LINE, 1,282, 32%), low complexity sequences (1,111, 27.7%) and others (1,615, 40.3%). Details on the classifications are reported in Supplementary Section 3.7.

REVIEWERS' COMMENTS

Reviewer #1 (Remarks to the Author):

The authors have answered my questions and clarified my concerns, and I wish to congratulate them on their work. I have no further comments.

Sincerely,

Wouter De Coster

Reviewer #2 (Remarks to the Author):

All of my issues were addressed.

Reviewer #3 (Remarks to the Author):

I would like to thank the authors for taking the time to consider my comments and make the necessary revisions to your paper. Your effort to address the concerns raised in my review is greatly appreciated. The changes and updates you have made have improved the quality and clarity of your work.

Your attention to detail and commitment to presenting high-quality research is evident in the updated manuscript.

Once again, thank you for taking the time to incorporate my suggestions into your work. I look forward to reading the final version of your paper.